# Epidemiology, Staging and Management of Prostate Cancer

**DOI:** 10.3390/medsci8030028

**Published:** 2020-07-20

**Authors:** Adam Barsouk, Sandeep Anand Padala, Anusha Vakiti, Azeem Mohammed, Kalyan Saginala, Krishna Chaitanya Thandra, Prashanth Rawla, Alexander Barsouk

**Affiliations:** 1Department of Hematology-Oncology, Hillman Cancer Center, University of Pittsburgh, Pittsburgh, PA 15232, USA; adambarsouk@comcast.net; 2Department of Medicine, Nephrology, Medical College of Georgia, Augusta University, Augusta, GA 30912, USA; azmohammed@augusta.edu; 3Department of Medicine, Hematology-Oncology, Medical College of Georgia, Augusta University, Augusta, GA 30912, USA; avakiti@augusta.edu; 4Plains Regional Medical Group Internal Medicine, Clovis, NM 88101, USA; drsaginala@gmail.com; 5Department of Pulmonary and Critical Care Medicine, Sentara Virginia Beach General Hospital, Virginia Beach, VA 23454, USA; kc_thandra@yahoo.com; 6Department of Internal Medicine, Sovah Health, Martinsville, VA 24112, USA; rawlap@gmail.com; 7Hematology-Oncology, Allegheny Health Network, Pittsburgh, PA 15212, USA; alexbarsouk@comcast.net

**Keywords:** prostate cancer, epidemiology, etiology, incidence, mortality, survival, risk factors, prevention, staging, treatment

## Abstract

Prostate cancer is the second most common and fifth most aggressive neoplasm among men worldwide. It is particularly incident in high human development index (HDI) nations, with an estimated one in seven men in the US receiving a prostate cancer diagnosis in their lifetime. A rapid rise and then fall in prostate cancer incidence in the US and Europe corresponded to the implementation of widespread prostate specific antigen (PSA) testing in 1986 and then subsequent fall from favor due to high rates of false positives, overdiagnosis, and overtreatment (as many as 20–50% of men diagnosed could have remained asymptomatic in their lifetimes). Though few risk factors have been characterized, the best known include race (men of African descent are at higher risk), genetics (e.g., BRCA1/2 mutations), and obesity. The Gleason scoring system is used for histopathological staging and is combined with clinical staging for prognosis and treatment. National guidelines have grown more conservative over the past decades in management, recommending watchful waiting and observation in older men with low to intermediate risk disease. Among higher risk patients, prostatectomy (robotic is preferred) and/or external beam radiotherapy is the most common interventions, followed by ADT maintenance. Following progression on androgen deprivation therapy (ADT) (known as castration-resistance), next generation endocrine therapies like enzalutamide, often in combination with cytotoxic agent docetaxel, are standard of care. Other promising treatments include Radium-223 for bone metastases, pembrolizumab for programmed death ligand-1 (PDL1) and microsatellite instability (MSI) high disease, and poly ADP ribose polymerase (PARP) inhibitors for those with mutations in homologous recombination (most commonly BRCA2).

## 1. Introduction

One in seven men in the US [1] and one in 25 worldwide [2] is likely to receive a prostate cancer diagnosis within his lifetime. The prostate is a walnut-sized reproductive organ located caudal to the bladder and cranial to the penis. As an exocrine gland, the prostate produces seminal fluid that nourishes and facilitates the transport of sperm (produced in the testicles) during and after ejaculation. The alkalinity of prostatic fluid helps to protect the sperm in the acidic environment of the vagina [3]. The prostate naturally enlarges with age, known as benign prostatic hyperplasia (BPH), causing symptoms (e.g., urinary frequency due to bladder compression) in one-third of men over 60 and about half over 80 [4]. BPH has been characterized by some as a stepping stone to prostate cancer tumorigenesis due to common histopathology and molecular drivers, but their precise relationship remains hotly contested [5,6]. Prostate cancer has become especially common in the developed world, though many point to the ready accessibility of PSA testing as a chief driver of rising incidence, over-diagnosis, and over-treatment of prostate cancer [7,8]. Recent developments in the ever-evolving landscape of prostate cancer epidemiology, staging and management guidelines are summarized below.

## 2. Epidemiology

### 2.1. Incidence

According to recent GLOBOCAN data, there were an estimated 1,276,000 cases of prostate cancer worldwide in 2018, which equals an incidence rate of 29.3/100,000 men, and comprises 7.1% of all cancer diagnoses (Figure 1) [9]. This incidence makes prostate cancer the second most common neoplasm among men worldwide, comprising 13.5% of all male cancer diagnoses (after lung cancer, which accounts for 14.5%). The cumulative global lifetime risk among men of prostate cancer is 3.73% [2].

Throughout the Americas, much of Europe and Africa, and Oceania, prostate cancer is the most common cancer diagnosis among men. However, due to a propensity of lung cancer in populous Asian nations, namely China, prostate cancer remains the global leader in incidence. The regions with the highest prostate cancer incidence are Australia/New Zealand (at 86.4/100,000) and Northern and Western Europe. The lowest rates were observed in South Central (5.0/100,000) and South-Eastern Asia. Prostate cancer is significantly more common in high HDI nations (such as the US, UK and Japan), with an age-standardized incidence rate (ASR) of 37.5/100,000, which nearly matches lung cancer at 40.4/100,000 (among men). Meanwhile, in low-medium HDI nations (such as China, India and Brazil), the ASR is 6.3 (compared to 11.8 for lung). It remains contested as to how much of a role broader access to PSA testing may play in this discrepancy [2].

In the United States, 192,000 cases of prostate cancer are estimated in 2020, accounting for 10.6% of all new cancer diagnoses. This incidence has steadily decreased over the past decades, from a peak of 237.6/100,000 in 1992 to a value of 108.0/100,000 today. The lifetime risk in the US is approximately 12.1%, almost 4-times that of the global risk [1].

There were 48,487 new annual prostate cancer cases in the UK, accounting for 13% of total cancer diagnoses from 2015–2017. This represents a 41% increase in 1990 [10]. Among 16 African countries, the incidence of prostate cancer was 22.0 (95% CI: 19.93–23.97) and a median incidence rate of 19.5 per 100,000 population [11]. Among 169 studies, it was observed that men from disadvantaged areas had consistently lower rates of PSA testing and the prostate cancer incidence, advanced disease with a higher mortality rate [12].

The introduction of PSA testing in 1986 is hypothesized to have led to a rapid rise in prostate cancer incidence and decline in incidence since, may be due to the recent implementation of more conservative PSA testing recommendations [8].

### 2.2. Mortality

An estimated 359,000 men worldwide perished from prostate cancer in 2018, comprising 3.8% of all cancer deaths. Prostate cancer accounted for 6.7% of all cancer deaths in men, which made it the 5th most common cause of cancer mortality in men (Figure 2) [9]. In 46 of the 185 studied nations, prostate cancer was the number one cause of cancer death in men (as compared to 105 nations where it was the leading site of cancer incidence). The regions with the greatest rates of prostate cancer mortality include Sub-Saharan Africa and the Caribbean, with Barbados having the rates mortality worldwide [2].

While prostate cancer diagnosis is over 3-fold more common in high HDI nations, prostate cancer mortality in high HDI nations is much closer to that of the rest of the world (8.0 vs. 6.3/100,000 men), indicating improved survival [2].

In the United States, 33,000 men are estimated to die of prostate cancer in 2020, accounting for 5.5% of all cancer deaths. The current mortality is 18.9/100,000, down from a peak of 39.3/100,000 in 1993 [1]. There were 11,714 annual deaths from prostate cancer in the UK from 2015–2017, accounting for 7% of cancer mortality (greater than the 5.5% in the US). This represents an 18% increase in mortality rates since 1970, as compared to the US, where mortality has remained largely unchanged [10].

### 2.3. Survival

In the developed world, prostate cancer survival rates have steadily improved over the past decades. The most recently reported 5-year survival in the US, as of 2016, is 97.8%, which is a dramatic improvement since 1975, when the rate was 66.9% [1]. Most of this improvement occurred in the 1980s, when widespread PSA testing resulted in earlier discovery and resection of many asymptomatic prostate cancer cases [8]. The 5-year survival in the UK from 2013 to 2017 was 86.6%, as compared to 97.8% in the US [10].

## 3. Risk Factors

### 3.1. Age

Prostate cancer is common among elderly males [2]. Due to increasing PSA testing and life expectancy, elderly men are being diagnosed with prostate cancer [13]. It has been observed that the risk increases after 40 years in African Americans or in patients with positive family history and after 50 years among White men without any family history of prostate cancer [14].

### 3.2. Race

Prostate cancer seems to have a strong ethnic association. Men of African descent are at an increased risk of the disease. In the US, African Americans are more likely to be diagnosed with prostate cancer and 2.5 times more likely to die of the disease [15]. Black race was associated with increased risk of prostate cancer mortality according to the Surveillance, Epidemiology, and End Results (SEER) US population registry [16] but not in the Veterans Affair’s System or National Cancer Institute-sponsored randomized clinical trials (Figure 3 and Figure 4) [16]. Nonetheless, comorbid conditions such as heart disease and social determinants of health such as poverty and racial prejudice may all play a role in worse outcomes for African American men [17]. Men in Sub-Saharan Africa were over 5-fold more likely to die of prostate cancer than African Americans in the US, although part of this discrepancy can be chalked up to lack of access to healthcare [18].

### 3.3. Genetics

Several hereditary mutations, most notably the BRCA2 gene, have been associated with an increased risk of prostate cancer. BRCA1 and 2 are homologous recombination proteins, mutations which are more common among Ashkenazi Jewish populations and frequently associated with an increased risk of breast and ovarian cancer. Nevertheless, BRCA2 has also been shown to increase the risk of prostate cancer by 8.6-fold in men over 65 (as well as pancreatic cancer) or a 2.64-fold increase among all men. BRCA1 has been shown to confer a smaller risk [19,20]. BRCA2 carriers are recommended for prostate cancer screening beginning at age 40 [7]. Poly-ADP Ribose Polymerase (PARP) inhibitors, used in BRCA mutant breast and ovarian cancers due to synthetic lethality, are currently undergoing clinical trials for BRCA mutant prostate cancer patients [21]. Other genes associated with prostate cancer include ATM (odds ratio (OR) = 2.18); Homebox B13 (OR = 3.23); Lynch Syndrome, a mutation in mismatch repair genes (OR = 4.87); and CHEK2 (OR = 1.98) [22].

### 3.4. Family History

Both genetic and environmental factors predispose to the development of prostate cancer. The risk of developing prostate cancer increases in patients with a history of multiple family members and earlier age of diagnosis, and it increases by two to three-fold in those with first degree relatives with prostate cancer [23,24,25]. About 5% of the risk is associated with inherited genetic background, and about 20% of the patients report a positive family history with increased risk for high-penetrance genes [26,27,28,29]. The hereditary prostate cancer (HPC) genes linked to prostate cancer include HPC1 (located at 1q24–25) and HPCX (located at Xq27–28) [30,31]. Mutations in the RNASEL gene (located on 1q25) are also linked to the development of prostate cancer [32]. RNASEL gene plays a role in the innate immunity, thus combating viruses and regulation of apoptotic cell death [33]. A clinically more aggressive variant of prostate cancer has been seen in subsets of HPC with BRCA1 and *2* mutations [34].

### 3.5. Obesity

Obesity is the best characterized among the few known modifiable risk factors for prostate cancer. Obesity has been implicated in dysregulation of the insulin axis, inflammatory cytokine signaling, and induction of DNA-damaging oxidative stress, increasing the risk of several neoplasms, including breast, colorectal, and prostate cancers [35,36,37]. Category I obese men were found to be at a 20% increased risk of prostate cancer mortality, while category II obese men were at a 34% increased risk [38]. Obesity was also associated with an increased risk of more aggressive disease, treatment failure, and prostate-cancer specific mortality, along with mortality from comorbid condition [39].

### 3.6. Smoking

Tobacco smoking has been shown to increase prostate cancer incidence and mortality, likely because the mutagens present in cigarettes promote tumorigenesis of prostatic epithelial cells. In a study of middle-age men, smokers were shown to have an OR of 1.4 for prostate cancer diagnosis, while smokers with a greater than 40-pack year history had an OR of 1.6 [40]. A separate study found the risk of prostate cancer mortality was likewise 1.6× greater among smokers [41]. Smoking cessation appeared to have a favorable effect on prostate cancer incidence and mortality, with a greater effect as time since cessation increased [40,41].

### 3.7. Diet

Diets high in calcium and dairy and low in alpha-tocopherol and selenium have been suggested to increase the risk of prostate cancer. In an extensive meta-analysis by The Prostate Cancer SLR, 4 of 15 studies analyzed reported a significant positive association between dairy and PC incidence, with 5 of 15 studies finding a 7% increased risk per 400 g of dairy each day. Thirteen of 15 studies found a significantly increased risk with calcium consumption, with 15 of 16 studies finding a 5% increased risk per 400 mg daily. Calcium is known to downregulate the active form of Vitamin D3 (1,25-dihydroxy vitamin D3), resulting in proliferation of prostatic cells. In that same meta-analysis, low intake of alpha-tocopherol (an isomer of Vitamin E) and Selenium were found to statistically increase PC risk in 2/11 and 2/10 studies analyzed, respectively. Beta-carotene, a precursor of Vitamin A, was found to have no significant impact on PC risk according to the meta-analysis [42].

### 3.8. Other Risk Factors

A meta-analysis of 53 studies found that vasectomies, an elective contraceptive surgical procedure that ligates the vas deferens, was not associated with an increased risk of PC [43]. Meanwhile, another meta-analysis found that a history of prostatitis, or chronic inflammation of the prostate, significantly increased overall PC risk in the general population but not among African Americans [44]. Non-steroidal anti-inflammatory agents inhibit cyclo-oxygenase (COX) activity and are used for a wide array of cardiovascular and immune indications. A meta-analysis found that the anti-inflammatory effect of NSAIDs, especially those that are COX-2 selective, could be protective against PC, particularly among men with a history of prostatitis [45]. 5-α reductase inhibitors (5-ARIs) are used to treat benign prostate hyperplasia (BPH) and have been suggested to prevent the development of PC. However, two randomized control trials found that 5-ARIs may actually increase the risk of aggressive, high-grade PC, leading the FDA to issue a “black-box” warning against these drugs. Other studies since have disputed the increased risk of high-grade PC and suggested a possible role in early-stage PC prevention, but the FDA warning remains in place [46,47].

### 3.9. Diagnosis and Staging

#### 3.9.1. Diagnosis

PSA testing was first approved in 1986 as a cheap and effective means of detecting asymptomatic prostate cancer cases. Routine PSA testing of all adult men led to a rapid increase in prostate cancer incidence in the developed world, which has since decreased, likely in part due to more restrictive PSA testing guidelines. While PSA is expected to normally rise with age (due to BPH), a PSA level of 4–10 ng/mL is considered borderline and portends an approximately 25% risk of prostate cancer; a PSA > 10 is associated with a greater than 50% risk [8].

PSA testing significantly improved prostate cancer survival rates due to earlier detection of disease, allowing for resection and local treatment prior to metastasis [8]. However, routine PSA testing of low-risk individuals has also been shown to have downsides. One trial found that over 10 years of testing, 15% of men received a false positive result, causing them to undergo unnecessary and invasive diagnostic procedures like biopsy [48]. Trials suggest that 20–50% of men diagnosed with prostate cancer due to PSA testing may be “over-diagnosed,” meaning their disease would have remained asymptomatic throughout their lifetimes and the treatment they underwent was thus unnecessary [49].

The American Cancer Society recommends PSA testing for average risk men starting at 50, African Americans at 45, and at 40 for high-risk men with an early age first-degree relative. However, the US Preventive Task Force does not recommend PSA screenings on account of the harms outweighing the benefits [48].

Positive PSA testing is commonly followed up with a biopsy. Prostate cancer biopsies have an approximately 1% chance of resulting in hospitalization due to sequela of the procedure [36]. Trans-rectal ultrasound and/or magnetic resonance imaging (MRI) is often used to guide the biopsy and quantify prostate size for diagnosis. If the biopsy is positive, the cancer is graded by Gleason Score.

Recently approved molecular biomarkers such as Decipher, Oncotype DX Prostate, Prolaris, or ProMark are recommended by the NCCN guidelines for low-risk disease but not by the American Society of Clinical Oncology (ASCO) or the American Urological Association (AUA).

#### 3.9.2. Grading and Staging

Gleason score is the leading histopathological scoring system for prostate cancer. Grade 1 corresponds to well-differentiated, low grade dysplastic tissue, while Grade 5 corresponds to the most abnormal, dysplastic tissue. Scores from the most representative pathological samples are summated for a total score. While scores between 2–10 are possible, scores below 6 are rarely found. Gleason score is then used to stratify prostate cancers into low-grade (<6), intermediate grade (7) and high-grade (8–10) disease. Grade groups correspond to Gleason scores more specifically than these categories. Risk stratification is then made based on grade group, PSA, and clinical stage (Tumor Node Metastasis staging) [50].

Several imaging modalities are used to search for metastases and stage metastatic prostate cancer, such as multiparametric MRI [51], bone scan (technetium-99 m diphosphonate scintigraphy), and computer tomography (CT). The AUA only recommends these imaging modalities for high risk or unfavorable intermediate risk patients prostate cancer patients [52]. Multiparametric-magnetic resonance imaging (mp-MRI) has shown promising results in diagnosis, localization, risk stratification and staging and possible treatment options for the prostate cancer. T2 weighed imaging remains the mainstay for the diagnosis of prostate cancer [50]. Recent literature has suggested the beneficial role of PSA, and mp-MRI, followed by targeted biopsy of the lesion, is a substitute to transrectal ultrasound guided biopsy. There is a significant advantage for risk stratification and in turn avoiding an unnecessary biopsy [53].

Transrectal transducer is the commonest imaging modality for prostate gland, and it also assists in obtaining the biopsy and for brachytherapy. It cannot reliably provide information about staging or differentiating between benign and malignant lesions. CT also offers minimal information about the imaging, staging, and the extent of its spread [54,55]. An MRI should be performed before a biopsy to avoid an unnecessary prostate biopsy.

PROMIS (Prostate MR imaging study) is aiming to introduce mp-MRI before the prostate biopsy thus avoiding unnecessary procedures and obtaining samples directly from the suspicious areas. To overcome the accuracy issues associated with the transrectal approach, it is using transperineal biopsy [56]. Mp-MRI before the biopsy might avoid unnecessary procedures by a quarter and improves recognition of clinically significant lesions, thus avoiding over-diagnosis of clinically insignificant lesions as cancer [57]. Compared to transperineal prostate biopsy, the incidence of urinary tract infections was 5.4 times more common in transrectal guided biopsies [58].

## 4. Treatment and Management

Definitive treatment for localized disease includes radical prostatectomy and/or radiotherapy. During radical prostatectomy, urologists will often survey and remove sentinel lymph nodes to assess for tumor spread. Lymph nodes can also be assessed via needle biopsy. Among low-risk or intermediate-risk patients, especially those with a life-expectancy of <5 years, observation or watchful waiting is recommended over treatment for non-aggressive prostate cancer [59]. Robotic or laparoscopic prostatectomy is associated with equivalent efficacy, less blood loss and fewer intra-operative complications [60]. Non-nerve sparing, non-robotic prostatectomies, especially among men 65 and older, are associated with a higher risk of erectile dysfunction [61]. Complications of robotic or laparoscopic or open Prostatectomy include incontinence of urine, erectile dysfunction, urethral strictures, and an increased risk of inguinal hernias with overall mortality less than 1%. The probability of erectile dysfunction is highly dependent on the age and the type of surgery (nerve sparing versus non-nerve sparing) [54,62,63].

External beam radiotherapy can be used as monotherapy for low-risk or favorable intermediate risk patients, as well as an adjuvant treatment for high risk patients following prostatectomy [64].

High-dose rate brachytherapy involves use of high dose radiation to target the cancer, thus sparing the surrounding organs. It is indicated as a monotherapy or in combination with external beam ration and also as a salvage option following prior external beam therapy [65]. For patients with low to intermediate risk disease, low-dose rate brachytherapy alone be offered, whereas a combination of low- or high-dose rate brachytherapy along with external beam radiation (with or without androgen deprivation therapy) should be offered for patients with high-intermediate risk of prostate cancer [66].

Radiotherapy can also be combined with ADT for high risk patients, though this increases the risk of systemic side effects and loss of sexual function [67]. Whole gland cryosurgery is used as an alternative to prostatectomy and radiotherapy in patients with life expectancy of over 10 years but who cannot tolerate these due to comorbidities. HIFU and Focal therapy have not been proven equally effective for localized disease [52].

ADT has become the standard of care for initial castration-sensitive prostate cancer treatment (both local and metastatic) thanks to the STAMPEDE trial initiated in 2005 [68]. Gonadotropin-releasing hormone agonists and antagonists have largely replaced surgical orchiectomy as the primary form of ADT in the developed world and have been shown to reduce symptoms like bone pain and slow tumor progression.

Patients who have disease progression (both metastatic and nonmetastatic) while on effective ADT (testosterone < 50 ng/dL) are considered castration resistant (CRPC) and require next-generation endocrine agents to suppress their cancer. These therapies can include those that inhibit androgen biosynthesis, like abiraterone [69], or the newer and more commonly used agents that interfere with androgenic stimulation of prostate cancer cells, like enzalutamide [70,71], apalutamide [72], or darolutamide [73]. The cytotoxic agents docetaxel and cabazitaxel, which over-stabilize microtubules and disrupt cell division, are also commonly employed. These agents all show significant survival improvement when added to ADT upon progression [70,71,72]. Few trials have compared these agents head-to-head, meaning there is little consensus on proper sequencing.

Enzalutamide significantly prolonged the survival of men with metastatic castration-resistant prostate cancer after chemotherapy [74]. Though the sample size was small, Schmid et al. have shown modest clinical benefit with consecutive use of enzalutamide and abiraterone after taxane-based chemotherapy [75]. Thus, sequence therapy alternating between chemotherapy and antihormonal drugs might be a more promising approach in metastatic castration resistant prostate cancer treatment [75].

Pembrolizumab, a PDL-1 inhibitor, was recently approved for later line treatment for patients with high PDL-1 expression and microsatellite instability (MSI) across cancer types [76]. PDL-1 is manipulated by tumor cells to evade immune destruction, and monoclonal antibodies against the PDL-1 receptor on T-cells can negate this effect. In prostate cancer, pembrolizumab has been shown to durable anti-tumor activity in those who have progressed on endocrine therapies (like enzalutamide) and docetaxel [77]. Other options for CRPC include sipuleucel-T, another immunotherapy indicated for minimally symptomatic patients [78], and Radium-223, indicated for patients with only bone metastases (no visceral metastases) [79].

PARP inhibitors are currently being evaluated in clinical trials as front-line therapy for all patients with mutations in homologous recombination genes, such as BRCA1/2, ATM, CHEK2, and other rare mutations [21,80].

Metastatic involvement is the most important prognostic factor in CRPC, with overall survival significantly higher for those with lymph-node only disease (31.6 months) than for those with metastases to bone (21.3), lung (19.4), or liver (13.5) [81].

## 5. Conclusions

Prostate cancer is the second most common and fifth most aggressive neoplasm among men worldwide. In the Americas and Western Europe, it is the most common cancer, with 1 in 7 men likely to receive a diagnosis within their lifetimes. Risk factors for the disease include race (men of African descent are at higher risk), genetics (e.g., BRCA1/2 mutations), and obesity. Implementation of PSA testing in 1986 led to a rapid rise in prostate cancer incidence, but incidence has since fallen as routine PSA testing has fallen somewhat out of favor due to high false positive rates and overdiagnosis (as many as 20–50% of men diagnosed may be unnecessarily treated). Disease is staged clinically as well as pathologically using the Gleason scoring system. National guidelines have become more conservative with treatment recommendations, recommending for watchful waiting and observation in older men with low to intermediate risk disease. For higher risk disease, prostatectomy (robotic may be safer) and/or external beam radiotherapy are the most common treatments, followed by long-term ADT. When prostate cancer progressed on ADT, next generation endocrine therapies like enzalutamide, often in combination with cytotoxic agent docetaxel, have become the standard of care. Other promising options include pembrolizumab for PDL1 and MSI-high disease, Radium-223 for bone metastases, and PARP inhibitors for those with mutations in homologous recombination (most commonly BRCA2).

## Figures and Tables

**Figure 1 medsci-08-00028-f001:**
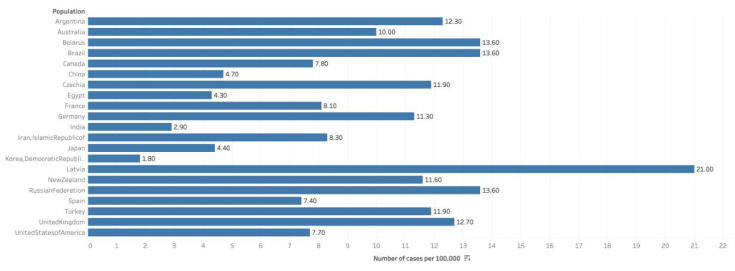
Graph showing Estimated cumulative risk of incidence in 2018, prostate cancer, males, ages 0–74. Data obtained from GLOBOCAN 2018 [9].

**Figure 2 medsci-08-00028-f002:**
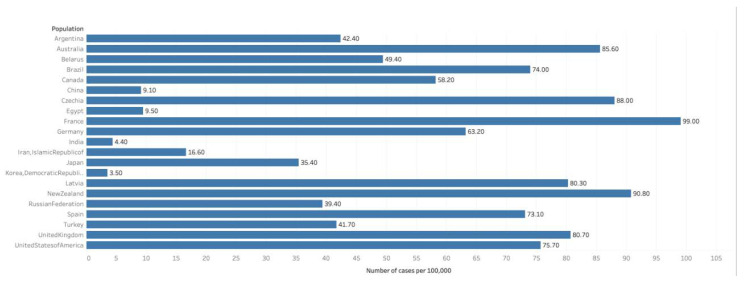
Graph showing Estimated cumulative risk of mortality in 2018, prostate cancer, males, ages 0–74. Data obtained from GLOBOCAN 2018 [9].

**Figure 3 medsci-08-00028-f003:**
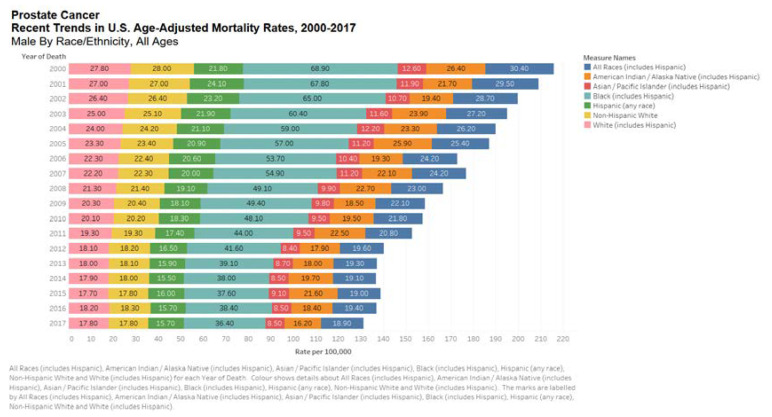
Bar chart showing United States age-adjusted mortality rates, 2000–2017 by all races. Data source: SEER*Explorer [16].

**Figure 4 medsci-08-00028-f004:**
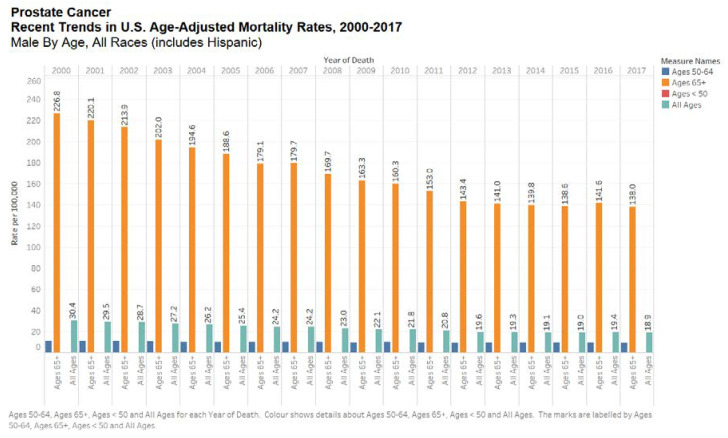
Bar chart showing recent trends in the United States age-adjusted mortality rates, 2000–2017 by all ages. Data source: SEER*Explorer [11].

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
