# Peer review of "Epidemiology, Staging and Management of Prostate Cancer"

_medsci, 2020, doi:10.3390/medsci8030028_

Round 1
Reviewer 1 Report
In this review, the authors firstly give an epidemiological overview of prostate cancer. But this overview is solely based on one resource, GLOBOCAN. As a review article, it will be more insightful to have multiple references. And it is helpful to give some examples of those high or low HDI nations. It was described that Australia/New Zealand has the highest incidence, but New Zealand data is missing in the figure1 and 2.
The whole genetic risk factor paragraph needs to be re-formated, and title is missing.
There are also some grammar errors: line193, line247 PD-L1, line249.
Author Response
First Reviewer
- In this review, the authors firstly give an epidemiological overview of prostate cancer. But this overview is solely based on one resource, GLOBOCAN. As a review article, it will be more insightful to have multiple references.
As for the first recommendation, we have already cited both GLOBOCAN and the SEER statistics. We have also included the statistics from the Cancer Research UK.
Incidence: There were 48,487 new annual prostate cancer cases in the UK, accounting for 13% of total cancer diagnoses from 2015-2017. This represents a 41% increase in 1990.
Mortality: There were 11,714 annual deaths from prostate cancer in the UK from 2015-2017, accounting for 7% of cancer mortality (greater than the 5.5% in the US). This represents an 18% increase in mortality rates since 1970, as compared to the US, where mortality has remained largely unchanged.
Survival: The 5-year survival in the UK from 2013 to 2017 was 86.6%, as compared to 97.8% in the US.
https://www.cancerresearchuk.org/health-professional/cancer-statistics/statistics-by-cancer-type/prostate-cancer/incidence#heading-Two
Among 16 African countries, the incidence of prostate cancer was 22.0 (95% CI: 19.93–23.97) and a median incidence rate of 19.5 per 100,000 population.
https://journals.plos.org/plosone/article?id=10.1371/journal.pone.0153496
Among 169 studies, it was observed that, men from disadvantaged areas had consistently lower rates of PSA testing and the prostate cancer incidence, advanced disease with a higher mortality rate.
https://www.frontiersin.org/articles/10.3389/fonc.2019.00238/full
- And it is helpful to give some examples of those high or low HDI nations.
high HDI nations (such as the US, UK and Japan), with an age-standardized incidence rate (ASR) of 37.5/100,000, which nearly matches lung cancer at 40.4/100,000 (among men). Meanwhile, in low-medium HDI nations (such as China, India and Brazil)
- The whole genetic risk factor paragraph needs to be re-formated, and title is missing.
Genetics
Several hereditary mutations, most notably the BRCA2 gene, have been associated with an increased risk of prostate cancer. BRCA1 and 2 are homologous recombination proteins, mutations in which are more common among Ashkenazi Jewish populations and frequently associated with an increased risk of breast and ovarian cancer. Nevertheless, BRCA2 has also been shown to increase the risk of prostate cancer by 8.6-fold in men over 65 (as well as pancreatic cancer), or a 2.64-fold increase among all men. BRCA1 has been shown to confer a smaller risk14,15. BRCA2 carriers are recommended for prostate cancer screening beginning at age 407. Poly-ADP Ribose Polymerase (PARP) inhibitors, used in BRCA mutant breast and ovarian cancers due to synthetic lethality, are currently undergoing clinical trials for BRCA mutant prostate cancer patients16. Other genes associated with prostate cancer include ATM (odds ratio (OR)=2.18), Homebox B13 (OR= 3.23), Lynch Syndrome, a mutation in mismatch repair genes (OR=4.87) and CHEK2 (OR= 1.98)17.
- Corrections in Line 193, 247 and 249
Corrections are done - Changed to PDL-1, and at 40 for high-risk men with an early age first-degree relative.
- Included New Zealand in Incidence and Mortality
©
Reviewer 2 Report
I have indicated suggested changes in the attachment. Software used is pdf expert.
Age not listed as a risk factor
The role of mpMRI not given due credence
transperineal prostate biopsy not mentioned
brachytherapy not mentioned as an option for organ-confined prostate cancer
enzulatamide is used instead of docetaxel, not in combination with it.
Would have liked more on side effects of therapies, for example incontinence with surgery

Author Response
- Age not listed as a risk factor:
Age
Prostate cancer is common among elderly males (2). Due to increasing PSA testing and the life expectancy, elderly men are being diagnosed with prostate cancer (13). Rawla, P. (2019). Epidemiology of Prostate Cancer. World Journal Of Oncology. 2019:10(2);63-89.). It has been observed that the risk increases after 40 years in African Americans or in patients with positive family history and after 50 years among White men without any family history of prostate cancer (14- Cancer Stat Facts: Prostate Cancer [Internet]. SEER, 2018. Available from: https://seer.cancer.gov/statfacts/html/prost.html).
- Role of mpMRI
Multiparametric-magnetic resonance imaging (mp-MRI) has shown promising results in diagnosis, localization, risk stratification and staging and possible treatment options for the prostate cancer. T2 weighed imaging remains the mainstay for the diagnosis of prostate cancer diagnosis.
Recent literature has suggested the beneficial role of PSA and mp-MRI, followed by targeted biopsy of the lesion, is a substitute to transrectal ultrasound guided biopsy. There is a significant advantage for risk stratification and inturn avoiding an unnecessary biopsy.
Ghai S, Haider MA. Multiparametric-MRI in diagnosis of prostate cancer. Indian J Urol. 2015;31(3):194-201. doi:10.4103/0970-1591.159606.
- Transperineal prostate biopsy not mentioned
Transrectal transducer is the commonest imaging modality for prostate gland, and it also assists in obtaining the biopsy and for brachytherapy. It cannot reliably provide information about staging or differentiating between benign and malignant lesions. CT also offers minimal information about the imaging, staging and the extent of its spread.
Leslie SW, Soon-Sutton TL, et al. Prostate Cancer. [Updated 2019 Oct 8]. In: StatPearls [Internet]. Treasure Island (FL): StatPearls Publishing; 2020 Jan-. Available from: https://www.ncbi.nlm.nih.gov/books/NBK470550/
Hedge JV, Mulkern RV, Panych LP, et al. Multiparametric MRI of prostate cancer: an update on state-of-the-art techniques and their performance in detecting and localizing prostate cancer. J Magn Reson Imaging. 2013;37(5):1035-1054. doi:10.1002/jmri.23860.
An MRI should be performed before a biopsy to avoid an unnecessary prostate biopsy. Xiang et al have concluded that, the diagnostic accuracy is same for both transperineal and transrectal prostate biopsy, the transperineal route had higher risk of pain, but had a lower risk of fever and rectal bleeding.
Xiang J, Yan H, Li J, Wang X, Chen H, Zheng X. Transperineal versus transrectal prostate biopsy in the diagnosis of prostate cancer: a systematic review and meta-analysis. World J Surg Oncol. 2019;17(1):31. Published 2019 Feb 13. doi:10.1186/s12957-019-1573-0
Compared to transperineal prostate biopsy, the incidence of urinary tract infections were 5.4 times more common in transrectal guided biopsies.
Skouteris VM, Crawford ED, Mouraviev V, et al. Transrectal Ultrasound-guided Versus Transperineal Mapping Prostate Biopsy: Complication Comparison. Rev Urol. 2018;20(1):19-25. doi:10.3909/riu0785
- Brachytherapy
High dose rate brachytherapy involves use of high dose radiation to target the cancer, thus sparing the surrounding organs. It is indicated as a monotherapy, or in combination with external beam ration, and also as a salvage option following prior external beam therapy.
Mendez LC, Morton GC. High dose-rate brachytherapy in the treatment of prostate cancer. Transl Androl Urol. 2018;7(3):357-370. doi:10.21037/tau.2017.12.08
For patients with low to intermediate risk disease, low-dose rate brachytherapy alone be offered, whereas a combination of low or high dose rate brachytherapy along with external beam radiation (with or without androgen deprivation therapy should be offered for patients high-intermediate risk prostate cancer.
Hannoun-Lévi JM. Brachytherapy for prostate cancer: Present and future. Cancer Radiother. 2017;21(6-7):469-472. doi:10.1016/j.canrad.2017.06.009
Enzalutamide significantly prolonged the survival of men with metastatic castration-resistant prostate cancer after chemotherapy.
Scher HI, Fizazi K, Saad F, et al. Increased survival with enzalutamide in prostate cancer after chemotherapy. N Engl J Med. 2012;367(13):1187-1197. doi:10.1056/NEJMoa1207506
Though the sample size was small, Schmid et al, have shown modest clinical benefit with consecutive use of enzalutamide and abiraterone after taxane-based chemotherapy. Thus, sequence therapy alternating between chemotherapy and antihormonal drugs might be a more promising approach in mCRPC treatment.
Schmid, S.C., Geith, A., Böker, A. et al. Enzalutamide After Docetaxel and Abiraterone Therapy in Metastatic Castration-Resistant Prostate Cancer. Adv Ther 31, 234–241 (2014). https://doi.org/10.1007/s12325-014-0092-1
- Complications
Complications of Radical Prostatectomy include incontinence of urine, erectile dysfunction, urethral strictures, and an increased risk of inguinal hernias with overall mortality less than 1%. The probability of erectile dysfunction is highly dependent on the age and the type of surgery (nerve sparing versus non-nerve sparing).
Leslie SW, Soon-Sutton TL, Sajjad H, et al. Prostate Cancer. [Updated 2019 Oct 8]. In: StatPearls [Internet]. Treasure Island (FL): StatPearls Publishing; 2020 Jan-. Available from: https://www.ncbi.nlm.nih.gov/books/NBK470550/
Moncada I, López I, Ascencios J, Krishnappa P, Subirá D. Complications of robot assisted radical prostatectomy. Arch. Esp. Urol. 2019 Apr;72(3):266-276.
Bratu O, Oprea I, Marcu D, Spinu D, Niculae A, Geavlete B, Mischianu D. Erectile dysfunction post-radical prostatectomy - a challenge for both patient and physician. J Med Life. 2017 Jan-Mar;10(1):13-18

Round 2
Reviewer 2 Report
Better than first version.
I would like to see specific mention of the PROMIS UK study about mpMRI.
I am very surprised that transrectal biopsy is found equivalent to transperineal biopsy in prostate cancer diagnoses. You cannot access the anterior prostate adequately via transrectal ultrasound. In London, UK virtually all prostate biopsies in all centres are transperineal. I question the authors' reference.
Although mentioned, I would like a section specifically on family history. For example, a first degree relative with prostate cancer doubles your risk. And a family history of breast cancer increases your risk. This is separate to the BRCA story.
Where radical prostatectomy is mentioned, I would prefer the authors to list robotic/laparoscopic/open, not just robotic as this is not accessible to much of the world.
Author Response
I would like to see specific mention of the PROMIS UK study about mpMRI
I am very surprised that transrectal biopsy is found equivalent to transperineal biopsy in prostate cancer diagnoses. You cannot access the anterior prostate adequately via transrectal ultrasound. In London, UK virtually all prostate biopsies in all centres are transperineal. I question the authors' reference.
PROMIS (Prostate MR imaging study) is aiming to introduce mp-MRI before the prostate biopsy thus avoiding unnecessary procedures and also in obtaining samples directly from the suspicious areas. To overcome the accuracy issues associated with transrectal approach, it is using transperineal biopsy56. Mp-MRI before the biopsy might avoid unnecessary procedures by a quarter and improves in recognizing in clinically significant lesions, thus avoiding over-diagnosis of clinically insignificant lesions as a cancer57.
- El-Shater Bosaily A, Parker C, Brown LC, et al. PROMIS--Prostate MR imaging study: A paired validating cohort study evaluating the role of multi-parametric MRI in men with clinical suspicion of prostate cancer. Contemp Clin Trials. 2015;42:26-40. doi:10.1016/j.cct.2015.02.008.
- Ahmed HU, El-Shater Bosaily, Brown LC, et al. Diagnostic accuracy of multi-parametric MRI and TRUS biopsy in prostate cancer (PROMIS): a paired validating confirmatory study. The Lancet. 2017;389(10071):815-
The reference citing equal efficacy between transrectal and transperineal has been removed.
Although mentioned, I would like a section specifically on family history. For example, a first degree relative with prostate cancer doubles your risk. And a family history of breast cancer increases your risk. This is separate to the BRCA story.
Family history
Both genetic and environmental factors predispose to the development of prostate cancer. The risk of developing prostate cancer increases in patients with a history of multiple family members and earlier age of diagnosis, and it increases by two to three-fold in those with first degree relatives with prostate cancer23-25. About 5% of the risk is associated with inherited genetic background and about 20% of the patients report a positive family history with increased risk for high-penetrance genes26-29. The hereditary prostate cancer (HPC) genes linked to prostate cancer include HPC1 (located at 1q24-31) and HPCX (located at Xq27-28)30,31. Mutations in the RNASEL gene (located on 1q25) are also linked to the development of prostate cancer32. RNASEL gene plays a role in the innate immunity, thus combating viruses and regulation of apoptotic cell death33. A clinically more aggressive variant of prostate cancer has been seen in subsets of HPC with BRCA1 and 2 mutations34.
- Lichtenstein P, Holm NV, Verkasalo PK, et al. Environmental and heritable factors in the causation of cancer--analyses of cohorts of twins from Sweden, Denmark, and Finland. N Engl J Med. 2000;343:78–85
- Bruner DW, Moore D, Parlanti A, et al. Relative Risk of Prostate Cancer for Men With Affected Relatives: Systematic Review and Meta-Analysis. Int J Cancer. 2003;107:797–803.
- Zeegers MP, Jellema A, Ostrer H. Empiric risk of prostate carcinoma for relatives of patients with prostate carcinoma: a meta-analysis. Cancer. 2003;97:1894–1903.
- Gallagher RP, Fleshner N. Prostate cancer: 3. Individual risk factors. CMAJ. 1998;159(7):807-813.
- Carroll PR, Grossfeld GD, editors. Prostate cancer. Hamilton, London: Decker Inc.; 2002
- Ferris-i-Tortajada J, Garcia-i-Castell J, Berbel-Tornero O, et al. [Constitutional risk factors in prostate cancer]. Actas Urol Esp. 2011;35(5):282-288.
- Sridhar G, Masho SW, Adera T, et al. Association between family history of prostate cancer. JMH. 2010;7:45-54.
- Smith JR, Freije D, Carpten JD, et al. Major susceptibility locus for prostate cancer on chromosome 1 suggested by a genome-wide search. Science. 1996;274:1371–1374.
- Xu J, Meyers D, Freije D, et al. Evidence for a Prostate Cancer Susceptibility Locus on the X Chromosome. Nat Genet. 1998;20:175–179.
- Carpten J, Nupponen N, Isaacs S, et al. Germline mutations in the ribonuclease L gene in families showing linkage with HPC1. Nat Genet. 2002;30:181–184.
- Zhou A, Paranjape J, Brown TL, Nie H, Naik S, Dong B, Chang A, et al. Interferon action and apoptosis are defective in mice devoid of 2',5'-oligoadenylate-dependent RNase L. EMBO J. 1997;16(21):6355-6363.
- Erkko H, Xia B, Nikkila J, Schleutker J, Syrjakoski K, Mannermaa A, Kallioniemi A, et al. A recurrent mutation in PALB2 in Finnish cancer families. Nature. 2007;446(7133):316-319.
Where radical prostatectomy is mentioned, I would prefer the authors to list robotic/laparoscopic/open, not just robotic as this is not accessible to much of the world
Changed to => Complications of robotic or laparoscopic or open Prostatectomy include incontinence of urine, erectile dysfunction, urethral strictures, and an increased risk of inguinal hernias with overall mortality less than 1%. The probability of erectile dysfunction is highly dependent on the age and the type of surgery (nerve sparing versus non-nerve sparing)42,49,50